# Recent genome reduction of *Wolbachia* in *Drosophila recens* targets phage WO and narrows candidates for reproductive parasitism

Jason A. Metcalf[1], Minhee Jo[1], Sarah R. Bordenstein[1], John Jaenike[2] and Seth R. Bordenstein[1,3]

[1] Department of Biological Sciences, Vanderbilt University, Nashville, TN, USA
[2] Department of Biology, University of Rochester, Rochester, NY, USA
[3] Department of Pathology, Microbiology, and Immunology, Vanderbilt University, Nashville, TN, USA

Corresponding author
Seth R. Bordenstein,
s.bordenstein@vanderbilt.edu

## ABSTRACT

*Wolbachia* are maternally transmitted endosymbionts that often alter their arthropod hosts' biology to favor the success of infected females, and they may also serve as a speciation microbe driving reproductive isolation. Two of these host manipulations include killing males outright and reducing offspring survival when infected males mate with uninfected females, a phenomenon known as cytoplasmic incompatibility. Little is known about the mechanisms behind these phenotypes, but interestingly either effect can be caused by the same *Wolbachia* strain when infecting different hosts. For instance, *w*Rec causes cytoplasmic incompatibility in its native host *Drosophila recens* and male killing in *D. subquinaria*. The discovery of prophage WO elements in most arthropod *Wolbachia* has generated the hypothesis that WO may encode genes involved in these reproductive manipulations. However, PCR screens for the WO minor capsid gene indicated that *w*Rec lacks phage WO. Thus, *w*Rec seemed to provide an example where phage WO is not needed for *Wolbachia*-induced reproductive manipulation. To enable investigation of the mechanism of phenotype switching in different host backgrounds, and to examine the unexpected absence of phage WO, we sequenced the genome of *w*Rec. Analyses reveal that *w*Rec diverged from *w*Mel approximately 350,000 years ago, mainly by genome reduction in the phage regions. While it lost the minor capsid gene used in standard PCR screens for phage WO, it retained two regions encompassing 33 genes, several of which have previously been associated with reproductive parasitism. Thus, WO gene involvement in reproductive manipulation cannot be excluded and reliance on single gene PCR should not be used to rule out the presence of phage WO in *Wolbachia*. Additionally, the genome sequence for *w*Rec will enable transcriptomic and proteomic studies that may help elucidate the *Wolbachia* mechanisms of altered reproductive manipulations associated with host switching, perhaps among the 33 remaining phage genes.

## INTRODUCTION

*Wolbachia* are widespread obligate intracellular $\alpha$-proteobacteria that infect around 40% of arthropod species (*Zug & Hammerstein, 2012*) and 47% of filarial nematodes (*Ferri et al., 2011*). These infection frequencies, if extrapolated to the diversity and abundance of their hosts, make *Wolbachia* perhaps the most widespread endosymbiont in animals. To maximize its propagation in arthropods, the maternally inherited *Wolbachia* has evolved an assortment of mechanisms to distort its host's reproductive system in a manner that enhances the relative production of infected females. These mechanisms include feminization, parthenogenesis, male killing, and cytoplasmic incompatibility (CI), the most common phenotype and one that results in embryonic lethality when matings occur between infected males and uninfected females (*Werren, Baldo & Clark, 2008*). Females harboring the same *Wolbachia* strain, meanwhile, can successfully mate and produce infected offspring with either infected or uninfected males, giving these females a selective advantage in populations of mixed infection status.

Interestingly, some *Wolbachia* strains are multipotent and can induce more than one type of reproductive manipulation depending on the arthropod host it infects (*Fujii et al., 2001*; *Jaenike, 2007*). In one striking example, the *Wolbachia* strain *w*Rec causes CI in its native host, *Drosophila recens*, but when introgressed into a sibling species, *D. subquinaria*, it causes male killing (*Jaenike, 2007*). Moreover in a natural hybrid zone between these same two species, unidirectional CI plays a major role in reducing interbreeding and thus contributes to reproductive isolation between these species (*Jaenike et al., 2006*; *Shoemaker, Katju & Jaenike, 1999*). Even though the link between *Wolbachia* and CI has been known for over 40 years (*Yen & Barr, 1971*), the mechanisms by which *Wolbachia* accomplishes its reproductive manipulations remain unknown.

Despite the physical isolation resulting from its intracellular lifestyle, *Wolbachia* in arthropods are replete with mobile DNA (*Wu et al., 2004*) including a temperate bacteriophage named WO (*Kent & Bordenstein, 2010*; *Metcalf & Bordenstein, 2012*; *Masui et al., 2000*). It has been speculated that WO may be involved in *Wolbachia* reproductive manipulations due to the prevalence of ankyrin repeat genes in its genome (*Wu et al., 2004*), the pervasiveness of phage-encoded bacterial virulence factors (*Boyd, 2012*), and the frequent occurrence of phage WO in arthropod *Wolbachia* strains (*Gavotte et al., 2007*). However, evidence not supportive of this hypothesis includes the observations that CI is inconsistently associated with the presence of phage WO genes (*Sanogo, Eitam & Dobson, 2005*; *Saridaki et al., 2011*) and that the penetrance of CI in *Nasonia* wasps is negatively correlated with densities of phage WO virions, as predicted by the phage density model (*Bordenstein et al., 2006*; *Bordenstein & Bordenstein, 2011*). Interestingly, PCR screening for the WO minor capsid gene specified WO's absence in *w*Rec (*Bordenstein & Wernegreen, 2004*), even though its closest relatives have large amounts of prophage DNA (*Wu et al., 2004*; *Klasson et al., 2009*). Thus, the absence of phage WO in *w*Rec would be a critical example of a *Wolbachia* strain causing multiple reproductive phenotypes but lacking WO.

To investigate the apparent lack of prophage WO genes and alternative genetic mechanisms behind *w*Rec's diverse phenotypic influences, we sequenced the *w*Rec genome

using next-generation sequencing technology with partial finishing via Sanger sequencing. We determined that although *w*Rec lacks the WO minor capsid gene typically used in diagnostic screens, it does contain a number of prophage WO genes. Thus, the possibility that WO influences *Wolbachia* reproductive manipulations cannot be eliminated, and those WO genes present in *w*Rec offer a streamlined candidate list of the WO genes that could cause reproductive parasitism. Additionally, the availability of genomic information for a *Wolbachia* strain that is known to switch reproductive phenotypes will enable genomic, transcriptomic, and proteomic approaches to investigate the mechanisms behind these phenotypes.

## MATERIALS & METHODS

The *w*Rec genome sequencing reads and annotated contigs can be accessed from NCBI Bioproject PRJNA254527.

### *Wolbachia* strain relatedness

Multi-locus sequence typing (MLST) genes were concatenated and a Bayesian phylogeny was inferred as previously described (*Baldo et al., 2006*). Briefly, selected fragments of MLST genes (coxA, gatB, fbpA, ftsZ, and hcpA) from *Wolbachia* strains with complete or nearly complete genome sequences were obtained from GenBank or the sequencing group's online repositories, concatenated for a total length of 2,079 bp, and aligned with MUSCLE (*Edgar, 2004*). jModelTest 2 (*Darriba et al., 2012*) was used to determine the best model of evolution for the set of MLST haplotypes (GTR + I + G), and a Bayesian phylogeny was inferred using Mr. Bayes (*Ronquist et al., 2012*) with a chain length of 1,100,000, burn-in of 100,000, and subsampling frequency of 200.

### Genome sequencing and assembly

DNA was extracted from a pool of 10 female *Wolbachia* infected Pittsford strain *D. recens* flies using a Puregene DNA purification kit (Qiagen, Venlo, Limburg). Shotgun sequencing of the *w*Rec genome was conducted using an Illumina Hi-Seq (Vanderbilt Sequencing Core, Nashville, TN) with 100 bp paired end sequencing. Reads were filtered using five available *Wolbachia* genomes, *w*Bm (*Foster et al., 2005*), *w*Mel (*Wu et al., 2004*), *w*Ri (*Klasson et al., 2009*), *w*Oo (*Darby et al., 2012*), and *w*Pip (*Klasson et al., 2008*), by mapping reads to these genomes with length and similarity fractions of 0.5 and keeping all mapped reads, using CLC Genomics Workbench version 6.0.4 (CLC Inc, Aarhus, Denmark). A *de novo* assembly with a length fraction of 0.5 and similarity fraction of 0.8 was then performed on filtered reads. Sequencing of whole *w*Rec-infected *D. recens* females produced over 24 million reads, of which nearly 4% matched one or more previously sequenced *Wolbachia* genomes. *De novo* assembly of *Wolbachia*-filtered reads yielded 159 contigs. A *de novo* assembly of unfiltered reads was also performed and any contigs with a portion of its sequence matching contigs obtained from the filtered assembly were added to scaffolds in a search for novel genes. Separately, reads were mapped to the *w*Mel genome with length and similarity fractions of 0.5 producing a rough consensus sequence to guide assembly of the de novo contigs into scaffolds, which were further refined with Sanger

sequencing of PCR amplifications using primers designed to bind either end of putatively adjacent contigs to yield a final draft genome consisting of 43 scaffolds.

### Annotation and comparative genomics

The *w*Rec genome was annotated using MicroScope (*Vallenet et al., 2009*), supplemented with manual curation based on homology with *w*Mel. A comparison of gene-gene identity between *w*Mel and *w*Rec was performed with a reciprocal best BLAST as previously described (*Moreno-Hagelsieb & Latimer, 2008*). For whole-genome alignments and analyses, *w*Rec scaffolds were concatenated in the order in which the majority of their genes appear in *w*Mel. Whole-genome alignment was performed with Mauve (*Darling, Mau & Perna, 2010*) and a circular genome plot was created with DNAPlotter (*Carver et al., 2009*). Manual annotations, BLAST searches, and sequence manipulation were performed with either CLC Genomics Workbench or Geneious V5.5.6 (Biomatters Ltd., Auckland, New Zealand). Ka/Ks rates and ratios were calculated using either single gene or concatenated whole genome CDS alignments with any alignments shorter than 30 amino acids removed (*Buschiazzo et al., 2012*), using the program DnaSP (*Librado & Rozas, 2009*). Genomic synteny was assessed with the Cloud Virtual Resource (CloVR) comparative pipeline (*Angiuoli et al., 2011*) and Sybil synteny gradient viewer (*Riley et al., 2012*) using the Data Intensive Academic Grid (DIAG) at the University of Maryland. The number of phage and phage-associated genes in *Wolbachia* genomes was determined based on current GenBank annotations and includes genes in the phage-packaged eukaryotic association module (S Bordenstein, unpublished data).

## RESULTS

### Taxonomy of *w*Rec

Phylogenetic analysis based on the concatenated multilocus sequence typing (MLST) genes (*Baldo et al., 2006*) confirms several previous reports that the supergroup A strain *w*Rec is closely related to *w*Mel (*Baldo et al., 2006*; *Werren & Jaenike, 1995*; *Ioannidis et al., 2007*; *Gueguen, Onemola & Govind, 2012*), a widespread strain infecting *D. melanogaster* (Fig. 1). In addition, phylogenetic analyses of each individual MLST gene support the same relationship of *w*Mel as the closest sequenced relative to *w*Rec (data not shown). To date, all sequenced *Wolbachia* genomes in supergroups A and B (*Werren, Zhang & Guo, 1995*), including *w*Mel (*Wu et al., 2004*), have contained significant amounts of phage WO DNA. Thus the potential absence of WO in *w*Rec was unexpected and precipitated the genomic analysis described below.

### Genome features of *w*Rec with targeted reduction of prophage WO

Full sequencing statistics and an overview of *w*Rec genome features are listed in Table 1. *w*Rec scaffolds ($N = 43$) consisted of a total sequence length of 1,126,653 bp containing 1,271 protein coding sequences. 99.7% of all nucleotides in coding sequences shared between *w*Rec and *w*Mel were identical, indicating little divergence between these two closely related genomes despite occupying hosts that diverged >50 million years ago

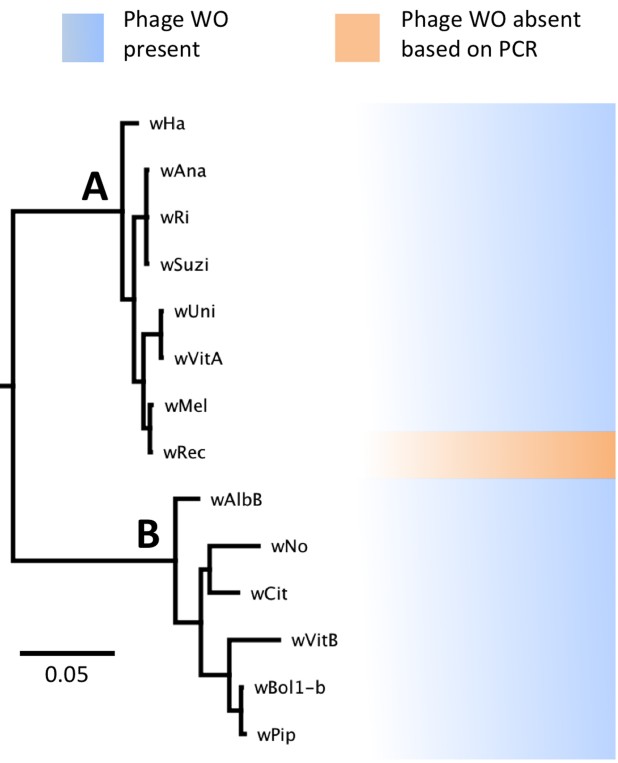

**Figure 1 WO phage is present in all sequenced supergroup (A) and (B) *Wolbachia* strains.** A Bayesian phylogeny based on the concatenated *Wolbachia* multi-locus sequence typing genes is shown, consisting of selected strains with partial or full genome sequences and *w*Rec. All branches had posterior probabilities of 99% or greater. While all previously sequenced *Wolbachia* strains in group (A) and (B) possess phage WO elements, *w*Rec (indicated with arrow), was formerly thought to be phage-free.

**Table 1  wRec sequencing and genome statistics.**

| | |
|---|---|
| Reads | 24,633,972 |
| *w*Rec reads | 955,730 (3.9%) |
| Contigs | 159 |
| Scaffolds | 43 |
| Average coverage | 76.5 |
| Genome size | >1,126,653 bp |
| GC content | 35.2% |
| CDS on scaffolds | 1271 |
| Average CDS length | 764 bp |
| Average intergenic length | 130 bp |
| Transfer RNA's | 34 |
| Ribosomal RNA's | 3 (23S, 16S, 5S) |
| Prophage regions | 2 |

(*Ross et al., 2003*). Based on a previously established rate of synonymous substitution in *Wolbachia* of 0.9% per million years (*Raychoudhury et al., 2009*), the genome-wide percentage of synonymous substitution (0.314%) between *w*Rec and *w*Mel puts their divergence at approximately 350,000 years ago. There were 2,009 single nucleotide polymorphisms (SNPs) between shared coding genes in *w*Mel and *w*Rec, and 599 (29.8%) of these SNPs were synonymous with an average $K_a/K_s$ ratio for each gene of 0.691. The vast majority of genes are highly conserved between *w*Rec and *w*Mel. More than 95% of orthologous gene pairs were 99% identical or greater and only ten gene pairs were less than 98% identical (Table 2). Most of these divergent genes code for hypothetical proteins and ankyrin repeat domain proteins. The *wsp* surface antigen, a known hypervariable sequence in *Wolbachia* (*Zhou, Rousset & O'Neil, 1998*), was also among the less conserved loci. All ten divergent genes contained insertions or deletions compared to *w*Mel in addition to one or more SNPs. Interestingly, four of these divergent genes, two coding for hypothetical proteins, an Ovarian Tumor (OTU)-like cysteine protease, and *wsp*, had $K_a/K_s$ ratios greater than one (Table 2), suggesting that they are evolving under positive selection, and the proteins they encode may be relevant to strain-specific host interactions. When these four genes were aligned to their homologs in *w*VitA, the closest relative of *w*Mel and *w*Rec, a roughly equal number of mutations in the OTU protease and *wsp* genes in each strain matched the sequence in *w*VitA. However, for the two hypothetical proteins WREC_0649 (WD_0722) and WREC_1268 (WD_1278), the *w*Mel alleles matched *w*VitA in a majority of cases (18 out of 25 nucleotides and 49 out of 56 nucleotides, respectively), suggesting that the *w*Mel variants were ancestral and that these *w*Rec alleles experienced lineage-specific positive selection during *D. recens* infection.

Interestingly, there were only two *w*Rec genes without nucleotide homology to genes in *w*Mel, even when contigs from a de novo assembly of raw host/*Wolbachia* reads were mapped to scaffolds in a search for additional genes. These two genes, WREC_0318 and WREC_0319, are hypothetical proteins with >95% nucleotide identity to sequences in two other *Wolbachia* strains, *w*Ri and *w*Ha. Meanwhile, *w*Rec lacked any homologs of 43 *w*Mel genes (Table S1), all but one of which are phage-related (phage genes discussed below). The single non-phage gene without homology in *w*Rec is WD_0032, which codes for a hypothetical protein with 96% similarity to the C-terminus of an ankyrin repeat-containing siRNA binding protein in *w*Ri. As is the case for many *Wolbachia* genomes, repetitive elements such as transposases and reverse transcriptases are abundant in *w*Rec and have hampered closing of the genome. 77 such repetitive genes were found in *w*Rec, and often appeared at the boundaries of scaffolds (Fig. 2). Although genomic rearrangement between the genomes cannot be completely assessed because the *w*Rec genome is not closed, genes in *w*Rec scaffolds were universally syntenic compared to *w*Mel (Fig. 3), with the exception of a 5 kb region containing WD_0042–WD_0051 (WREC_0853–WREC_0863), consisting of repetitive transposases, reverse transcriptases, hypothetical proteins, and pseudogenes. This region would have been located on the first *w*Rec scaffold if syntenic, but instead is on scaffold 31 (Fig. 3).

Table 2 **wMel genes with less than 98% nucleotide identity to their orthologs in wRec.** Genes with a $K_a/K_s$ ratio greater than one are highlighted.

| wMel locus | wRec locus | Function | Pairwise identity (%) | wRec length | wMel length | # SNPs | # Non-synonymous SNPs | $K_a/K_s$ ratio | Other changes |
|---|---|---|---|---|---|---|---|---|---|
| WD_0294 | WREC_0283 | Ankyrin repeat domain protein | 89.4 | 1,815 | 1,626 | 4 | 4 | – | 189 bp insertion |
| WD_0443 | WREC_0442 | OTU-like cysteine protease | 97.1 | 927 | 906 | 7 | 6 | 1.59 | 21 bp insertion |
| WD_0550 | WREC_0541 | Ankyrin repeat domain protein *TM domains | 87.4 | 789 | 990 | 2 | 2 | – | 99 bp deletion, C-terminal frameshift, alternate start/stop sites |
| WD_0722 | WREC_0649 | Hypothetical protein *TM domains | 92.0 | 462 | 450 | 25 | 21 | 4.25 | 9 bp insertion, 3 bp insertion |
| WD_0996 | WREC_0956 | Transposase | 89.1 | 744 | 801 | 1 | 0 | 0 | alternate start site, transposase insertion |
| WD_1007 | WREC_0973 | Hypothetical protein | 95.1 | 366 | 351 | 3 | 2 | 0.42 | 15 bp insertion |
| WD_1039 | WREC_1007 | Collagen triple helix repeat protein | 97.5 | 405 | 1,425 | 1 | 1 | – | 9 bp insertion, scaffold break |
| WD_1063 | WREC_1036 | Wsp surface antigen | 97.9 | 708 | 714 | 9 | 8 | 2.55 | 6 bp deletion |
| WD_1278 | WREC_1268 | Hypothetical protein *TM Domain | 92.1 | 2,604 | 2,766 | 56 | 51 | 6.07 | 162 bp deletion |
| WD_1298 | WREC_1289 | RpoD | 97.2 | 1,974 | 1,929 | 10 | 6 | 0.39 | 18 bp insertion, 27 bp insertion |

## Prophage WO relics in the genome

Whole-genome alignment of wRec and wMel revealed three major regions of genome reduction, with wRec lacking a large portion of both phage WO regions present in wMel as well as the entirety of the "Octomom" region (*Chrostek et al., 2013*) (Fig. 2), with only a bordering reverse transcriptase, WREC_0508 (wMel homolog WD_0506) present. Interestingly, although the minor capsid gene used in prior PCR surveys is absent, wRec does contain two major phage-related regions (Fig. 2). The first is a 19.2 kb region (WREC_0261–WREC_0285) across three scaffolds that is homologous to 21 contiguous genes of wMel WO-A (WD_0276–WD_0296). This region in wRec is syntenic and 99.4% identical to its homologous region in wMel, with two exceptions. The wRec homolog (WREC_0270/WREC_0274) of WD_0285, an ankyrin repeat protein, is fragmented by the insertion of two reverse transcriptases and a gap in the scaffolds, and there is an 189 bp insertion in WREC_0283 (WD_0294), another ankyrin repeat protein. If these two regions are included in the calculation of similarity, then the wRec WO-A phage region is 90.9% identical to the same region in wMel. The second wRec phage region contains

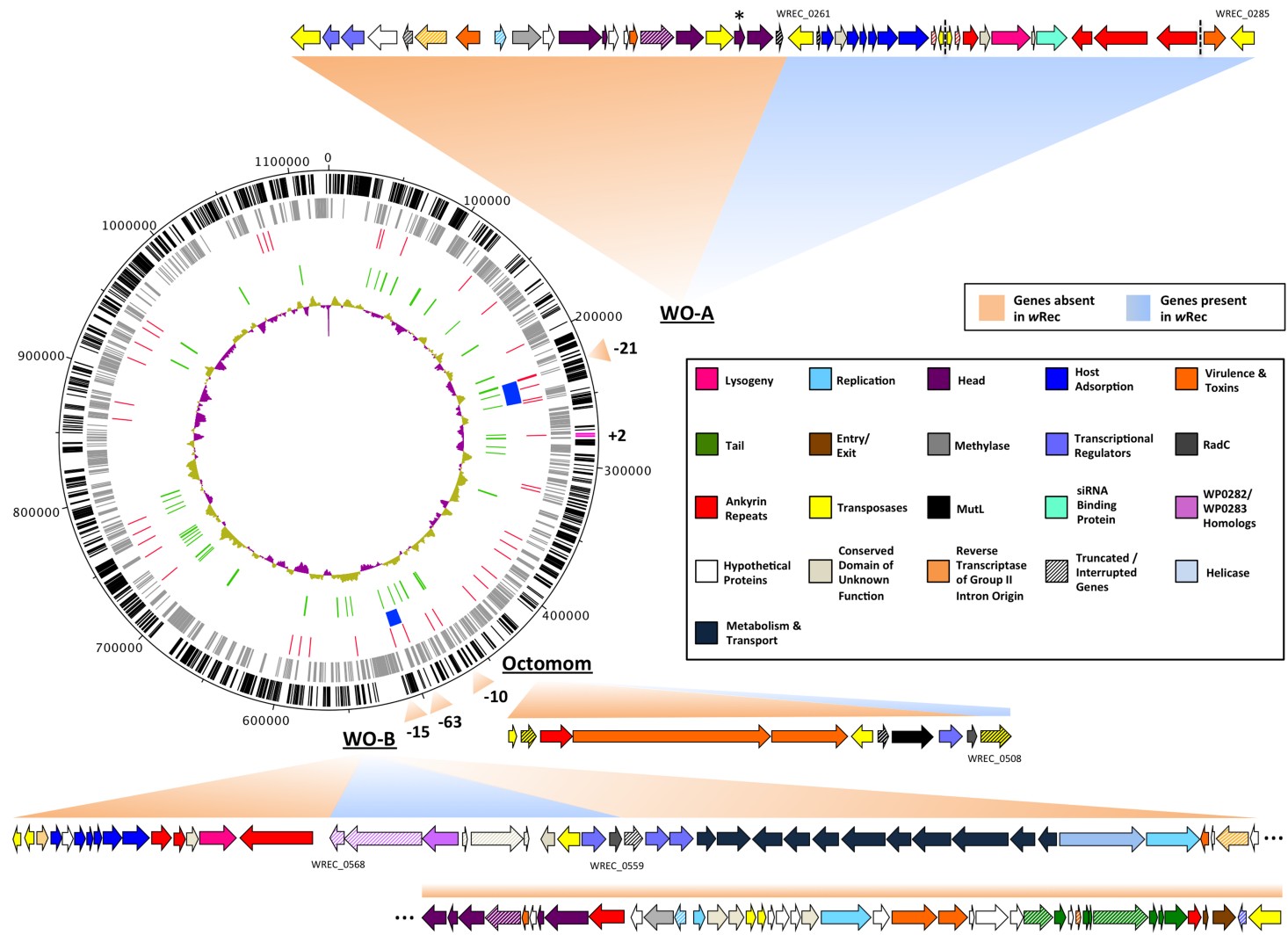

**Figure 2 *w*Rec genome comparison to *w*Mel.** *w*Rec scaffolds were concatenated in the order in which their genes appear in *w*Mel to produce the circular genome above. Major regions of loss or gain compared to *w*Mel are indicated outside the circle along with the number of genes involved. *w*Rec genome features are indicated within the circle plot as follows (from outside-in): 1 (black): CDS in forward direction, and (magenta) genes not found in wMel; 2 (grey): CDS in reverse direction; 3 (red): scaffold break points; 4 (blue): WO regions; 5 (green): transposases and reverse transcriptases; 6 (purple/gold): GC content variation from average. WO prophage and related regions are shown and genes are categorized by color according to their likely functions and presence/absence in *w*Rec. Locus tags for selected genes are indicated and dashed lines indicates breaks between scaffolds containing WO-A. The minor capsid gene of WO-A, which was used for prior PCR screens, is indicated with an asterisk.

11.3 kb and 7 genes (WREC_0559–WREC_0568) that are syntenic and homologous to part of WO-B in *w*Mel (WD_0625–WD_0632), with 99.5% pairwise identity. Two of these genes are interrupted by premature stop codons and the remaining fragments are annotated as smaller, separate genes. These genes include the orthologs of WD_0630, a hypothetical protein, which is split into three genes in *w*Rec (WREC_0563–WREC_0565), and the ortholog of WD_0632, which is split into the 3,096 bp gene WREC_0567 and 468bp gene WREC_0568. While the head region of WO appears to be absent in *w*Rec, the host adsorption module which is putatively involved in binding to the host surface during
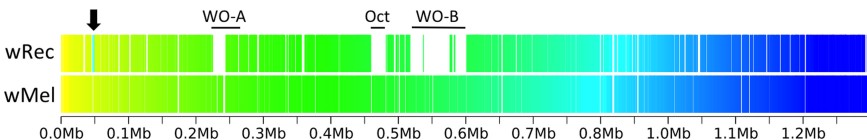

**Figure 3 Within-scaffold wRec synteny compared to wMel.** wRec scaffolds were concatenated in the order in which they appear in wMel and within-scaffold synteny was analyzed. Genes are graphed as tick marks colored on a gradient from yellow to blue from left to right with wMel as the reference genome and each wRec gene colored according to the location of its homolog in the wMel genome. White spaces in wRec alone indicate the absence of homologous genes or genes with multiple paralogs whose synteny cannot be established, while white spaces shared by both genomes indicate intergenic regions. A 5 kb region of rearrangement consisting of repetitive elements and hypothetical proteins is noted with an arrow, and phage-related regions are marked.

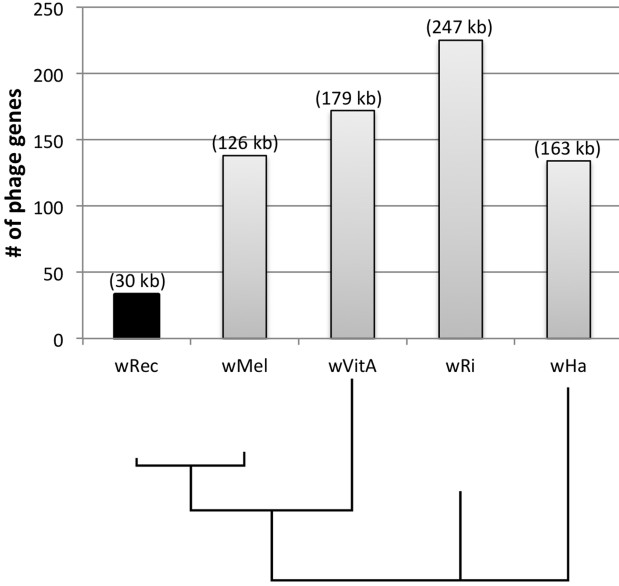

**Figure 4 Number of phage genes in wRec and its relatives.** The total number of prophage, phage-associated, and WO-like island genes in each *Wolbachia* genome is plotted above a Bayesian phylogeny of their MLST genes. The approximate total length of phage genes in each genome is noted above each bar.

phage infection is largely intact, as are a number of ankyrin repeat genes, a transcriptional regulator, and the homologs of WP_02082/WP_0283, two genes in wPip recently proposed as candidate mediators of CI (*Beckmann & Fallon, 2013*). In summary, the markedly reduced number of phage genes in wRec ($N = 33$) is the signature feature of the genome compared to its closest relatives, which possess anywhere from 134 (wHa) to 225 (wRi) phage or phage-associated genes (Fig. 4).

## DISCUSSION

### Divergence and genome reduction in *w*Rec

Genome analysis revealed that wMel and wRec are very closely related with an average of 99.7% nucleotide identity in coding regions shared by the two strains. We estimate that wRec and wMel diverged around 350,000 years ago. This estimate raises an interesting

biogeographical question: how could *Wolbachia* have been transferred at this time between the widely allopatric Nearctic *D. recens* and Afrotropical *D. melanogaster*? Perhaps a widespread *Drosophila*-generalist parasitoid played a role in vectoring this endosymbiont between host species, as parasitoid wasps have been previously demonstrated as vectors for *Wolbachia* transfer (*Heath et al., 1999*; *Vavre, Mouton & Pannebakker, 2009*). Molecular evidence suggests that the most recent *Wolbachia* sweep within *D. recens* occurred only 50,000 years ago, while *D. subquinaria* split from *D. recens* an estimated 600,000 years ago (*Shoemaker, Katju & Jaenike, 1999*). Thus, the divergence of *w*Mel and *w*Rec from their last common ancestor likely predated the most recent genetic sweep of *D. recens*, and *w*Rec infected *D. recens* after its incipient divergence from *D. subquinaria* (*Werren & Jaenike, 1995*; *Shoemaker, Katju & Jaenike, 1999*). Remarkably, these results suggest that *w*Rec may have contributed to reproductive isolation between these two species *prior* to the last glacial period 110,000–12,000 years ago, when their ranges are thought to have been allopatric (*Jaenike et al., 2006*). However, we note caution in interpreting the estimated divergence times as variability in mutation rates between bacterial lineages can skew the estimates.

The four *w*Rec genes evolving under positive selection are of particular interest as they may be potential mediators of *Wolbachia*-host interactions (Table 2). Indeed, *wsp* is known to be involved in pathogenicity and host interaction (*Uday & Puttaraju, 2012*) while OTU-like cysteine proteases have deubiquitinase activity facilitating the pathogenicity of intracellular pathogens and viruses (*Furtado et al., 2013*; *Makarova, Aravind & Koonin, 2000*). Although the function of the hypothetical proteins is unknown, the presence of transmembrane (TM) domains suggests interaction with the bacterial membrane and potentially its *Drosophila* host. Additionally, it has previously been speculated that the elevated rate of mitochondrial DNA evolution in *D. recens* was due to hitchhiking in association with a series of positive selection events in its resident *Wolbachia* (*Shoemaker et al., 2004*).

The major difference between the *w*Mel and *w*Rec genomes was the incipient genome reduction of WO prophage regions. Remaining phage WO genes in *w*Rec were often bordered or interrupted by transposases, suggesting that transposase activity may have been involved in the removal and degradation of major portions of WO genomes. Over 100 kb of genetic material, consisting mostly of phage-related genes, has likely been lost in *w*Rec. Unlike the prophages found in *w*Mel (*Wu et al., 2004*), all of *w*Rec's WO regions lacked the head genes thought to be necessary for mature virion formation (*Metcalf & Bordenstein, 2012*), including the *orf7* minor capsid protein used in previous PCR tests for WO (*Bordenstein & Wernegreen, 2004*). The lack of such head genes suggests that *w*Rec has lost the capacity to serve as a source of WO phage to infect other strains of *Wolbachia*. Future PCR screens may benefit from inclusion of more than one primer set, perhaps adding primers for a gene from the host adsorption module, which is highly conserved across WO prophages. However, it must be cautioned that the presence of multiple and variable degenerate WO haplotypes makes it impossible for any set of one or two primer pairs to detect all haplotypes.

Meanwhile, only 2,009 SNPs were present between the *w*Mel and *w*Rec genomes, indicating that gene deletion has been heavily favored over mutation. Such genome

reduction is common in obligate intracellular bacteria, where many genes are expendable due to relaxed selection and there is limited contact with novel gene pools (*Casadevall, 2008*). Given the predatory nature of intact WO phages (*Metcalf & Bordenstein, 2012*; *Bordenstein et al., 2006*; *Sanogo & Dobson, 2006*), it may have been evolutionarily advantageous for *w*Rec to eliminate the genes required for active phage production. It has been noted from TEM observations and quantitative studies that WO phage can lyse *Wolbachia*, resulting in an inverse correlation between bacterial and phage densities. Furthermore, because reproductive manipulations are dependent on a critical density of *Wolbachia*, high phage activity correlates with low expression of CI (*Bordenstein et al., 2006*). Since *w*Rec exhibits high levels of CI in *D. recens* (*Werren & Jaenike, 1995*), while *w*Mel shows lower levels of CI in *D. melanogaster* (*Yamada et al., 2007*), it is possible that *w*Rec experiences a higher selective pressure to suppress phage, preserve high bacterial densities, and maintain compatibility with its host's mating population. Thus, this interaction could be one possible explanation for the major loss of phage genes in *w*Rec that are preserved in *w*Mel.

Although *Wolbachia* has many more repetitive and mobile elements than most obligate intracellular bacteria (*Bordenstein & Reznikoff, 2005*) and frequently switches hosts on an evolutionary timescale (*Vavre et al., 1999*), it is worthwhile to note that there were only two genes in *w*Rec that were not present in *w*Mel. It is possible that these genes were lost in *w*Mel after divergence from its last common ancestor with *w*Rec.

## The phage WO hypothesis to explain reproductive parasitism

Because the Octomom region was completely absent in *w*Rec, it is unclear whether *w*Rec lost these genes after diverging from *w*Mel, or whether the genes were acquired by *w*Mel after divergence with their last common ancestor. Given that Octomom is not widespread in supergroup A *Wolbachia*, the latter possibility is likely. Moreover, although the function of Octomom in reproductive parasitism is unknown, it seems reasonable to conclude that the Octomom region is not needed for reproductive manipulations, as it is completely absent from *w*Rec. Additionally, given the association of Octomom with increased *Wolbachia* virulence, proliferation, and host viral protection (*Chrostek et al., 2013*), we would predict that *w*Rec would not possess these phenotypes, and may be a useful strain for confirming these associations.

It is intriguing that some WO genes are conserved in *w*Rec while others were lost. One explanation for their preservation in *w*Rec is that the remaining genes improve *Wolbachia* fitness. Indeed, prophage sequences code for advantageous virulence factors in a wide array of bacterial species (*Brussow, Canchaya & Hardt, 2004*). Because previous PCR surveys suggested *w*Rec did not possess phage WO, speculation that WO may be involved in *Wolbachia* reproductive manipulations has been largely disregarded (*Bordenstein & Wernegreen, 2004*). However, our sequencing shows that although the phage genomes are not complete, *w*Rec contains many phage-related genes including some that could be involved in CI and/or male-killing. These include at least four ankyrin repeat proteins, whose repetitive domain has been long thought to facilitate *Wolbachia*-eukaryote

interaction (*Iturbe-Ormaetxe et al., 2005*; *Siozios et al., 2013*). Additionally, several WO genes in *w*Rec are homologs of genes recently implicated in CI. WREC_0560 is a transcriptional regulator with 88.3% identity at the amino acid level to *wtrM* in *w*PipMol, which increases expression of an important regulator of meiosis in *Culex* mosquitos and is postulated to be a component of the molecular mechanisms of CI (*Pinto et al., 2013*). WREC_0566–WREC_0568 meanwhile, are homologous to WP_0282 and WP_0283, two genes in *w*Pip that have been implicated in CI due to presence in the proteome of *Wolbachia*-infected, fertilized mosquito spermathecae, along with their pattern of presence/absence in CI and non-CI strains (*Beckmann & Fallon, 2013*). Although the *w*Rec homolog of WP_0283 has been truncated by 427 bp, it has 99.8% nucleotide identity to the gene in *w*Mel (WD_0632) and an alternative reading frame enables the transcription of the remaining nucleotides in the same frame as the C-terminus of the homolog in *w*Mel. Whether any of these WO genes are actually involved in *Wolbachia* host manipulations remains unclear, especially since it is unknown whether the remnants of phage WO are transcribed by *w*Rec. However, the fact that these prophage regions are conserved suggests that they may have a role to play in the biology of *Wolbachia*.

## WO host adsorption genes

In addition to preservation of some potential reproductive manipulation mediators, prophage WO genes WREC_0263–WREC_0269 contain an intact host adsorption module that includes baseplate genes thought to be involved in the binding of WO to its bacterial host and insertion of phage DNA. Indeed, this host adsorption module is nearly universal in WO prophage, with very few degenerate phage haplotypes lacking these genes (*Kent et al., 2011*). Many intracellular bacteria, including *Wolbachia* (*Rances et al., 2008*; *Pichon et al., 2009*), possess a type IV secretion system that secretes effectors into the host as a common strategy to subvert host-cell functions (*Voth, Broederdorf & Graham, 2012*). A number of Gram-negative bacteria also possess a phage-like type VI secretion system (*Coulthurst, 2013*); these include several obligatory intracellular bacterial pathogens, such as *Anaplasma* and *Ehrlichia* (*Rikihisa & Lin, 2010*). Structural analyses have shown this type of secretion system bears a remarkable resemblance to the spike protein of phages (*Silverman et al., 2012*). Given these similarities, and the fact that the WO host adsorption module is almost universally present in sequenced arthropod *Wolbachia* (*Kent et al., 2011*), it is possible that *Wolbachia* may be using these genes to facilitate host-microbe interactions, as a way to inject CI factors, genes, or other host manipulation particles into its host.

## Future studies

The discovery of phage elements in *w*Rec opens up new questions. Additional experiments will be needed to determine whether any conserved phage genes are involved in *Wolbachia* manipulations of its host biology. In addition, we have seen that using single gene markers of phage WO is not diagnostic of its absence. Thus, unsequenced *Wolbachia* strains that were thought to be phage-free by PCR assays need reevaluation. Another question that remains is how a single *Wolbachia* causes multipotent reproductive manipulations in different host backgrounds. The availability of genomic sequence for a multipotent

*Wolbachia* strain will enable future transcriptomic and proteomic studies that could elucidate the genes involved in switching reproductive phenotypes.

## ACKNOWLEDGEMENTS

We are grateful to Jennifer Wisecaver for technical assistance and to two reviewers for their helpful feedback.

### Funding

This work was supported by NIH R01GM085163 (SRB), NSF DEB1046149 (SRB), NSF DEB-0542094 (JJ), and NIH T32GM07347 (Vanderbilt Medical Scientist Training Program). The funders had no role in study design, data collection and analysis, decision to publish, or preparation of the manuscript.

### Grant Disclosures

The following grant information was disclosed by the authors:
NIH: R01GM085163.
NSF: DEB1046149.
NSF: DEB-0542094.
NIH: T32GM07347.

### Competing Interests

The authors declare there are no competing interests.

### Author Contributions

- Jason A. Metcalf and Seth R. Bordenstein conceived and designed the experiments, performed the experiments, analyzed the data, contributed reagents/materials/analysis tools, wrote the paper, prepared figures and/or tables, reviewed drafts of the paper.
- Minhee Jo performed the experiments, analyzed the data, contributed reagents/materials/analysis tools, wrote the paper, reviewed drafts of the paper.
- Sarah R. Bordenstein performed the experiments, analyzed the data, contributed reagents/materials/analysis tools, wrote the paper, prepared figures and/or tables, reviewed drafts of the paper.
- John Jaenike contributed reagents/materials/analysis tools, wrote the paper, reviewed drafts of the paper.

### DNA Deposition

The following information was supplied regarding the deposition of DNA sequences:
NCBI Bioproject PRJNA254527.

### Supplemental Information

Supplemental information for this article can be found online at http://dx.doi.org/10.7717/peerj.529#supplemental-information.

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
