# Peer review of "Recent genome reduction of Wolbachia in Drosophila recens targets phage WO and narrows candidates for reproductive parasitism"

_PeerJ, doi:10.7717/peerj.529_

## Round 0.1 · original submission · Major Revisions

We thank you for your submission to PeerJ about the genome sequencing of a wolbachia strain from Drosophila recens. I give you great credit for this MS, however, you will need to address the issues raised by two reviewers - this will add value to this MS.

Reviewer 1 ·

Basic reporting

The paper by Metcalf et al report the genome sequencing of a wolbachia strain from Drosophila recens. The wRec genome shows a reduction of the prophage WO elements but not the lack of WO as previously postulated.
The paper is well written and adhering to all basic reporting fields. However on page 4 line 6 (….including a temperate bacteriophage named WO (Kent and Bordenstein, 2010, Metcalf and Bordenstein, 2012) the authors should cite the first paper reporting the bacteriophage WO in wolbachia (Masui et al (2000). J Mol Evol, 51:491–497) other than citing their reviews.

Experimental design

In the phylogenetic analysis the authors concatenated the MSLT genes and applied a model of substitution calculated on the genes as one. Since mrbayes allows to infer phylogenies on partitioned data sets in which each partition has its own model of substitution the authors should repeat the phylogenetic analysis calculating the best fit model for each genes and include these in mrbayes.
The authors filtered the sequence reads for wolbachia sequences from different strains but they do not mention how they performed this filtering and which parameters they have used.

Validity of the findings

I have no comment on the finding of the paper but I have a main comment on the interpretation of some of the finding. The authors have found four genes with a ratio of non-synonymous vs synonymous substitutions higher than one, interpreted as positive selection acting on these genes and that positive selection was acting on the wRec genes . While I do not disagreed on the positive selection acting on these genes I do not think the authors can say anything about on which lineage there is positive selection because they have compared just two wolbachia genomes. The authors should have taken in consideration these genes for more wolbachia strains and applied a likelihood ratio test. Test which would have allow them to see in which phylogenetic lineages there has been a significant w >1.
Finally in the results on page 9 and 10 line 23 and 1: …..with wRec having lost ….. as well as the entirety of the “Octomom” region. This phrase assume that both WO and Octomom has been lost on wRec but in the discussion the authors says that Octomom have more likely been acquired by wMel and not lost by wRec. I would rephrase this avoiding to reach conclusion about who gain and who lost in the results.

·

Basic reporting

A slight divergence from the standard layout in that the results and discussion are separate but I feel that this enhances the clarity of the manuscript.

My only suggested change is to add an 'of' into line 9 of page 12, so that this sentence reads -
"due to a hitchhiking in association with a series OF positive selection events in its resident Wolbachia"

Experimental design

No Comments

Validity of the findings

No Comments

Additional comments

A very well written and executed study. The discussion is very thorough and well reasoned. Overall, I really enjoyed reading this manuscript.

---

## Round 0.2 · accepted · Accept

Thanks for your MS to PeerJ. Congratulations, your MS has been accepted for publication. I hope you will keep this association with PeerJ and publish again in future.